

# Estimating robust melt factors and temperature thresholds for snow modelling across the Northern Hemisphere

Adrià Fontrodona-Bach[1,2], Bettina Schaefli[3], Ross Woods[4], and Joshua R Larsen[2,5]

[1]Institute of Science and Technology Austria, Klosterneuburg, Austria
[2]School of Geography, Earth and Environmental Sciences, University of Birmingham, Birmingham, United Kingdom
[3]Institute of Geography, GIUB, and Oeschger Centre for Climate Change Research, OCCR, University of Bern, Bern, Switzerland
[4]Department of Civil Engineering, University of Bristol, Bristol, United Kingdom
[5]Birmingham Institute for Forest Research (BIFoR), Birmingham, United Kingdom

**Correspondence:** Adrià Fontrodona-Bach (a.f.bach@bham.ac.uk)

**Abstract.** There are two important limitations to understanding large-scale impacts of environmental change on snow resources, 1) observational snow data at the point scale are highly limited, and 2) extrapolation using models can be challenging due to data availability and performance. This paper seeks to address these limitations using widely available climate network station data combined with a temperature-index snow model to derive estimates of mean snow water equivalent conditions across the Northern Hemisphere. Hydrological models commonly use very simple snow accumulation and melt models based on air temperature information, namely, a temperature threshold for snow accumulation as well as for snowmelt, and a melt factor. This utility emerges due to the simplicity, efficiency, and generally good performance of such models if sufficient calibration information is available. At scales beyond single gauged catchments, the estimation of the temperature thresholds and the melt factor has been difficult due to a lack of observations on snow accumulation and melt. Using a recently published Northern Hemisphere snow water equivalent dataset (NH-SWE) and co-located climate station observations of temperature and precipitation (5,560 sites across the Northern Hemisphere), this work provides the first large-scale and long-term (1950-2023) evaluation of a simple temperature index snow model and its parameters across a diverse range of snow climates. Our study reveals that the 0°C as snowfall-air temperature threshold captures most snowfall events, especially in cold climates, but risks missing 11% of snowfall events, especially in climates with regular near-freezing temperatures. Similarly, an air temperature threshold for snowmelt of 0°C reproduces well most daily snowmelt observations, but may lead to an earlier than observed onset of the melt season. Estimated melt factors converge towards 3-5 mm °C$^{-1}$ day$^{-1}$ for deeper snowpack climates (> 300 mm), but their estimation may be more challenging for colder climates with shallower snowpacks (< 300 mm), conditions where the inferred melt factors have much higher interannual variability. The temperature-index snow model performs consistently well across the available Northern Hemisphere data set for estimating long-term mean values of seasonal snow cover onset, snowmelt season onset, mean snow accumulation and snowmelt rates, but challenges may arise due to biases in temperature records or solid precipitation undercatch. This study provides valuable insights into temperature-threshold snowfall modelling and temperature-index melt modelling for applications across diverse climates and environments, and the results should help refine modelling approaches to enhance our understanding of snowpack responses to global warming.



## 1 Introduction

The sensitivity of snow accumulation and melt to warming temperatures is crucial for predicting global hydrological responses to climate change (Nijssen et al., 2001; Barnett et al., 2005; López-Moreno et al., 2020). Our understanding of snowpack sensitivity to rising temperatures can be improved by analysing trends in historical observations of snowpacks and climate variables from in-situ measurements (Fontrodona Bach et al., 2018; Luomaranta et al., 2019; Matiu et al., 2021) or from remote sensing (Bormann et al., 2018; Notarnicola, 2022), but these may be limited in time and space. Alternatively, spatio-
temporal snowpack dynamics can be analysed by simulating the response of snow accumulation and snowmelt under warming scenarios using models (Pomeroy et al., 2015; López-Moreno et al., 2021). However, model simulations are challenged by various sources of uncertainty (Oreskes et al., 1994), which may arise from natural variability (Willibald et al., 2020), parameter uncertainty (Günther et al., 2020), forcing data uncertainty (Günther et al., 2019; Terzago et al., 2020), the ability of models to represent certain processes (Cho et al., 2022), the choice of model configurations (Essery et al., 2013), or even subjective
modelling decisions (Melsen et al., 2019). Two crucial aspects to model development and model applications are the reliable formulation of assumptions, the reliable estimation of model parameters and the evaluation of model simulations against actual observations.

In the case of snow accumulation modelling, accurate estimation of the rainfall-snowfall partitioning is crucial (Behrangi et al., 2018). Most snowfall events occur below 0°C (Rohrer and Braun, 1994), but in alpine climates the actual zero degree
isotherm may lie 300-400 metres above the elevation at which snow starts to accumulate at the surface (Fabry and Zawadzki, 1995). Rain-snow partitioning is often better captured by the wet-bulb temperature, which depends on relative humidity and surface pressure (Jennings et al., 2018b). Inclusion of wet-bulb temperature to estimate the rain-snow partitioning has shown improved snow and streamflow simulations (Tobin et al., 2012; Zhang et al., 2015; Harpold et al., 2017; Jennings and Molotch, 2019; Wang et al., 2019), but these meteorological measurements generally have fair lower availability than (dry bulb) air
temperature. In practice, most contemporary studies derive the rainfall-snowfall threshold from precipitation phase observations or set it to some generally accepted value, which may differ from 0°C. A rainfall-snowfall separation temperature range between 0°C and 2°C is often used for hydrological modelling over mountain areas (Tobin et al., 2012; Bormann et al., 2014), and a wide range of studies use a fixed threshold of 0°C or 1°C even for large-scale applications (Berghuijs et al., 2014; Follum et al., 2019; Hou et al., 2023; Bonsoms et al., 2024).

There are two main approaches to modelling snowmelt, 1) solving the full energy balance, or 2) using simpler approaches that relate snowmelt directly to more readily available climate variables such as temperature and incoming radiation. The first approach is physics-based but computationally expensive and requires much more meteorological forcing data (e.g. relative humidity, wind speed, incoming and outgoing shortwave and longwave radiation, and often information on the snowpack itself) (Bartelt and Lehning, 2002; Lehning et al., 2002b, a; Magnusson et al., 2017). Although such models may be more
easily transferable and rely less on calibration, some physical parameters may not be well constrained and thus introduce considerable uncertainty in model simulations (Günther et al., 2020). Furthermore, applying this approach at large spatial



scales, beyond hillslopes or small catchments, comes with a reduction in forcing data resolution and, therefore a reduction in model reliability and performance (Magnusson et al., 2019).

Contrasting with the requirements of physics-based models, the simplest temperature-index snowmelt model (Lang and
Braun, 1990; Rango and Martinec, 1995; Hock, 2003) needs only air temperature data as forcing to simulate snowmelt. This approach relies on the key assumption that snowmelt can be directly related to air temperature, which generally holds well (Ohmura, 2001; Sicart et al., 2006). The melt factor (with units mm $°C^{-1}$ day$^{-1}$), is a parameter that captures how much melt is produced per degree of air temperature beyond a snowmelt temperature threshold. The melt factor has to be calibrated or assumed, and the temperature threshold above which snowmelt occurs can also be calibrated (Hock, 2003), but is often
assumed between -1°C and 1°C (Senese et al., 2014; Avanzi et al., 2022; Elias Chereque et al., 2024).

The interpretation and meaning of the melt factor are not trivial and fundamentally depend on the time and spatial scale over which they are estimated. Hock (2003) provides a comprehensive analysis of the variability of the melt factor and its physical basis. Generally, low melt factors indicate a low incoming shortwave radiation, a high albedo, a higher portion of melt driven by the sensible heat flux, or a combination of those. In these conditions, snowmelt is less sensitive to a change in temperature
(i.e. there is not much melt per degree of temperature), and therefore the melt factor is low. Conversely, high melt factors indicate a higher incoming shortwave radiation driving a larger part of the energy balance, which causes more snowmelt than what the relationship between melt and temperature alone can explain, and therefore the melt factor is high (Hock, 2003). The daily and seasonal variation of the melt factor with incoming shortwave radiation (Ismail et al., 2023) has motivated efforts to include shortwave radiation parameterisations of the melt factor (Pellicciotti et al., 2005; Magnusson et al., 2014), but this adds
complexity to the model. The melt factor also varies significantly spatially, across hydroclimatic regions, across land use or as a function of slope, aspect or sky view angles (Marsh et al., 2012). Reported melt factors for snow in the literature range between 0.5- and 20-$mm$ $°C^{-1}$ $day^{-1}$ across many different locations, but typical values range between 2- and 6- $mm$ $°C^{-1}$ $day^{-1}$ (Braithwaite, 1995; Kane et al., 1997; Lefebre et al., 2002; Hock, 2003; Braithwaite, 2008; Shea et al., 2009; Asaoka and Kominami, 2013).

Temperature-index models remain widely used for large-scale snow modelling due to their computational efficiency and ability to operate at high spatial resolutions (Rittger et al., 2016; Avanzi et al., 2023; Marty et al., 2025). This is relevant to better capture the high spatial variability of snow on the ground (López-Moreno et al., 2013, 2015), which is less well captured by Hemispheric-scale coarser gridded snow datasets (Mudryk et al., 2015). However, the temperature thresholds for rain-snow partitioning and for snowmelt initiation, and the melt factors used in temperature-index models are frequently either
assumed, spatially upscaled from limited point-scale observations, or calibrated at the catchment scale (Schaefli et al., 2005; Bogacki and Ismail, 2016; Riboust et al., 2019), without systematically evaluating how transferable these assumptions and parameterisations are across different climates. In particular, the melt factor has been estimated at a range of scales, from the the point, to catchment and glacier scales (Braithwaite, 1995; Lefebre et al., 2002; Shea et al., 2009; Ismail et al., 2023), but its spatial variability and performance across climates remain poorly constrained.

Given the widespread application and utility of temperature-index models, there is a need for a comprehensive assessment of temperature-index model assumptions, parameterisations, and performance across a wide range of snow climates, to better



understand the environmental conditions where these models are suitable and where they may fail. The availability of recently published station-based snow water equivalent (SWE) time series over the Northern Hemisphere, NH-SWE (Fontrodona-Bach et al., 2023b, a), and co-located climate station observations of temperature and precipitation, provides the possibility to gain

insights into the global variability of temperature-index melt model parameterisations, and offers the opportunity to test whether a temperature-index model can be robustly used across diverse snow climates.

Here, we use ground-based (station) observations of temperature and precipitation together with the NH-SWE time series to empirically derive and estimate the three key temperature-index model parameters of the classical degree-day model (the snowfall threshold, the snowmelt threshold, and the melt factor) across the Northern Hemisphere. We then use the temperature-

index model to simulate SWE time series using these point-scale observations of temperature and precipitation and evaluate the model's performance against the NH-SWE time series. This paper thus provides the first assessment of how a temperature-index model performs and how its parameters vary across climates, at the Northern Hemisphere scale.

## 2   Data

### 2.1   Temperature and precipitation data

We use daily time series of precipitation and temperature from the global historical climatology network dataset (GHCN-Daily, version 3.30-upd-2023080717) (Menne et al., 2012). Because of known biases in the temperature data within the SNOTEL dataset (Avanzi et al., 2014; Oyler et al., 2015), which is part of GHCN-Daily, we replace the SNOTEL data in GHCN-Daily with the Bias Corrected and Quality Controlled (BCQC) SNOTEL Data (Yan et al., 2018; Sun et al., 2019). For better coverage of the European continent, including the European Alps, we also use time series from the European Climate Assessment and

Dataset (ECA&D, last access: July 3rd 2023) (Klein Tank et al., 2002), and from the MeteoSwiss data portal IDAWEB (last access: August 17th 2023) (MeteoSwiss, 2023). If the mean daily temperature is unavailable, we use the daily minimum and maximum to compute the daily mean as the midpoint between the two. A total of 34,536 stations are initially available. We apply a simple gap-filling and quality control procedure (see Appendix A) and select stations with at least 7 years of gap-free data from 1950 to 2023, although the period available does not need to be continuous. A total of 14,881 stations with

temperature and precipitation time series are still available after these filters are applied.

### 2.2   Snow water equivalent data

We use the Northern Hemisphere daily snow water equivalent time series from the NH-SWE dataset (Fontrodona-Bach et al., 2023b, a). A total of 11,071 quality-controlled and gap-filled stations are initially available in the dataset. We match the locations of the stations in the NH-SWE dataset with the locations of the stations with temperature and precipitation time series. We select those that are separated by less than $0.01°$ latitude and $0.01°$ longitude (ca. 1 km distance) from each other,

with an elevation difference not larger than 10 metres, and which contain at least 7 years of overlapping gap-free data with the temperature and precipitation time series. A total of 5,560 stations from the NH-SWE dataset are matched with 4,076 stations





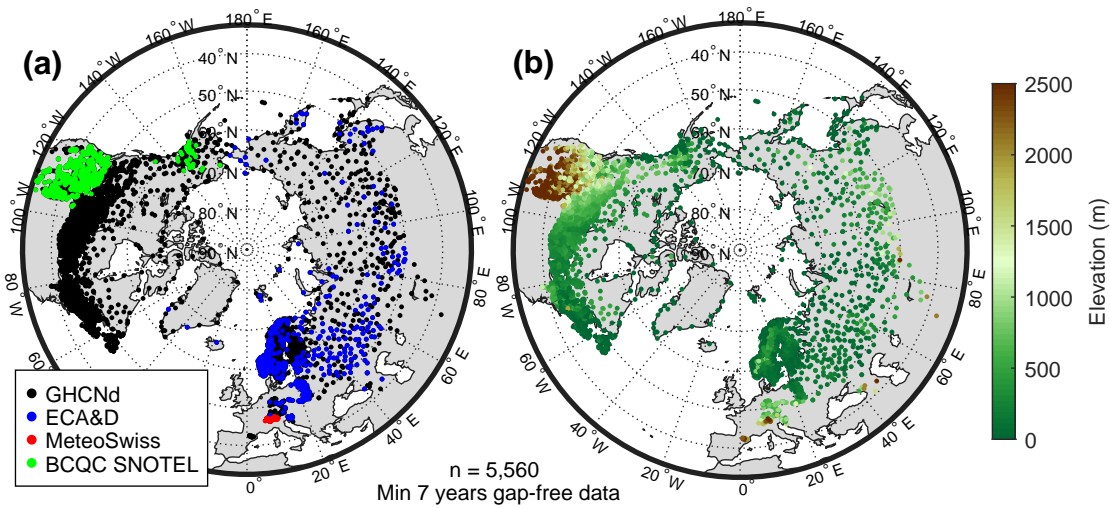

**Figure 1.** Matched NH-SWE stations with GHCNd/ECA&D/MeteoSwiss/BCQC-SNOTEL stations, and their elevation.

from the GHCN-d, 801 from the BCQC SNOTEL, 647 stations from ECA&D, and 36 stations from MeteoSwiss (Fig. 1). This results in a large range of stations across elevation, latitude and longitude gradients in the Northern Hemisphere (Fig. 1).

As the NH-SWE data contains only stations with at least 40 days of continuous snow cover on average, warmer climates and ephemeral snow climates are generally not included. The length of the remaining time series available shows the long-term nature of this study, with 50% of stations having at least 30 years of available data (Fig. B1).

## 3 Methods

### 3.1 Snow season and climate indices

We use the same snow season definitions as Fontrodona-Bach et al. (2023b), as well as two temperature-based indices from Woods (2009) to define the characteristic climate of each station (Table 1).

### 3.2 Temperature-index model

We simulate daily time series of snow water equivalent, SWE, using the daily time series of temperature and precipitation and a simple temperature-index model as follows:

$$\frac{dH_{\mathrm{SWE}}(t)}{dt} = A(t) - M(t), \tag{1}$$

where $H_{\mathrm{SWE}}(t)$ [mm] stands for SWE at time step $t$ [d], $A(t)$ [mm d$^{-1}$] is the snow accumulation at time step $t$ and $M(t)$ [mm d$^{-1}$] is the snowmelt at time step $t$. We use an explicit time-stepping scheme to solve Equation 1, i.e.:





$$H_{\mathrm{SWE},t} = H_{\mathrm{SWE},t-1} + \Delta(A_{t-1} + M_{t-1}), \qquad (2)$$

where $\Delta$ [d] is the time step length (in this study, daily), $A_{t-1}$ is the snow accumulation [mm] and $M_{t-1}$ the snowmelt [mm]
of the preceding day $t-1$ . $A_t$ is computed as follows:

$$A_t = \begin{cases} P_t & \text{if } T_t \leq T_a \\ 0 & \text{if } T_t > T_a \end{cases} \qquad (3)$$

where $P_t$ [mm d$^{-1}$] is the daily precipitation amount, $T_a$ [°C] is the threshold temperature for snow accumulation, and $T_t$
[°C] is the temperature on day $t$. $M_t$ is computed as follows:

$$M_t = \begin{cases} \alpha \times (T_t - T_m) & \text{if } T_t \geq T_m \\ 0 & \text{if } T_t < T_m \end{cases} \qquad (4)$$

where $\alpha$ [mm (°C d)$^{-1}$] is the melt factor, which we assume to be constant in time, and $T_m$ [°C] is the threshold temperature
for snowmelt. Snowmelt $M_t$ cannot exceed available snow, i.e. $M_t \leq H_{\mathrm{SWE},t} + A_t$.

**Table 1.** Definitions of snow season and climate indices used in this study.

| Term | Definition |
|---|---|
| Snow season | Longest period with continuous snow cover in a year. |
| Onset of accumulation season | Starting day of the snow season, lasting until peak SWE. |
| Peak snow water equivalent (peak SWE) | Highest snow water equivalent value in the snow season. |
| Onset of melt season | First day with a decrease in SWE after peak SWE. |
| Melt season | Period from onset of melt season to first day with zero SWE after peak SWE. |
| Snowmelt days | Days with a decrease in SWE. |
| Snowmelt rate | Sum of all SWE decreases in the melt season divided by the number of snowmelt days. |
| Mean annual temperature ($\overline{T}$) | Mean temperature over a full year. |
| Temperature amplitude ($\Delta_T$) | Difference between the warmest and coldest days of the annual cycle. |
| Melt season temperature | Mean temperature during the snowmelt season. |
| Mean annual snowfall | Sum of precipitation occurring at temperatures below $0$°C*. |

*This threshold is later varied for the temperature-index modelling but is kept constant here to obtain a mean snowfall climatology of a station.





## 3.3 Deriving and estimating temperature-index model parameters

We empirically derive and then estimate values for two parameters of the temperature-index model for all stations in our study: the snowfall temperature threshold ($T_a$) and the melt factor ($\alpha$). We also evaluate whether a 0°C threshold is a reasonable
assumption for the snowmelt temperature threshold ($T_m$).

As the SWE time series from the NH-SWE dataset are based on snow depth observations, an increase in SWE can be attributed to actual snow accumulation with high confidence. We can, therefore, empirically derive the snow accumulation temperature threshold by analysing the distribution of observed daily temperatures on days when there is snow accumulation. We set, for each station, the snowfall temperature threshold equal to the 80th percentile of the distribution (i.e., 80% of snow
accumulation days occur at or below this temperature threshold). This choice is a compromise between capturing the majority of snowfall events while avoiding excessive near-freezing precipitation to be classified as snow. Based on these snowfall thresholds empirically derived from observed data for each station, we build a simple multiple linear regression model that estimates the snow accumulation threshold based on the climate variables from Table 1 ($\overline{T}$ and $\Delta_T$). The parameter estimation model is shown in Section 4.1.1. We use the empirically derived and estimated parameters at each station to run the temperature-index
model simulations in Section 3.4.

Unlike for snow accumulation, the NH-SWE time series cannot confidently distinguish whether a decrease in SWE is due to snowmelt or due to an alternative loss mechanism such as sublimation or snow redistribution by wind, or due toe a model inaccuracy (Fontrodona-Bach et al., 2023b). The distribution of observed daily temperatures on days when there is a decrease in SWE can, thus, not be used to estimate a threshold temperature for snowmelt different from 0°C. Nevertheless, analysing the
distribution of temperatures at time of SWE decrease allows us to evaluate whether a 0°C threshold is a reasonable assumption, and is also useful to gain insights into temperature-index modelling limitations, as well as NH-SWE data limitations.

To empirically derive the daily melt factor (in mm °C$^{-1}$ day$^{-1}$) from the NH-SWE time series, we divide the daily absolute magnitude of SWE decrease by the corresponding observed daily temperature. Two observations are important here: 1) computed daily melt factors can become very large, or even infinite, if daily SWE decrease is small and temperature is close
to zero. 2) melt factors can become negative (physically not possible) if melt occurs on days with negative temperatures. We, therefore, remove melt factor values higher than 20 mm °C$^{-1}$ day$^{-1}$ (4% of all data), as these are rare in the literature (Asaoka and Kominami, 2013), and we also remove negative values (23% of all data). After removing these values, we define the annual melt factor at a site as the median of all daily melt factors in a year, and the melt season melt factor as the median of daily melt factors during the snowmelt season only (as defined in Table 1).
We build a simple multiple linear regression model that estimates the median melt factor in the melt season based on elevation, latitude and mean temperature, following a stepwise linear regression model that discarded $\Delta_T$ (Table 1) as a predictor for the melt factor. The parameter estimation model is shown in Section 4.1.3. We use the empirically derived and estimated parameters at each station to run the temperature-index model simulations in Section 3.4.





## 3.4 Temperature-index model simulations and performance

We simulate three daily SWE time series for each station, with three different parameter sets (Table 2). The first simulation is run with a common parameter set for all stations across the Northern Hemisphere, where the common snowfall temperature threshold is set to 0.5°C, and the common melt factor corresponds to the median of all estimated melt factors across the Northern Hemisphere. The second simulation is run with the empirically derived snowfall threshold and melt factor from observations at each station individually, as described in section 3.3. The third simulation is run with the estimated parameters

based on the multiple linear regression models built for snowfall threshold and melt factor (see sections 4.1.1 and 4.1.3). For all simulations, the threshold temperature for snowmelt is set to 0°C. An overview of the workflow to obtain the three parameter sets and to set up the model simulations is shown in Fig. 2, and the parameter values used are shown in Table 2.

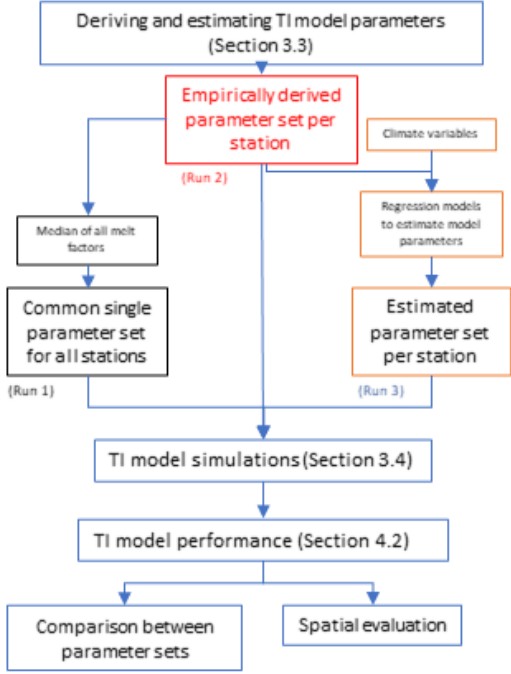

**Figure 2.** Flow diagram describing the temperature-index model simulations and the empirically derived and estimated parameter sets.

We evaluate the performance of the temperature-index model for important indicators of snow season dynamics, namely: accumulation season onset, snowmelt season onset, end of snow season, peak SWE, number of snowmelt days, and snowmelt

rate (Table 1). Timing errors are measured in days, with negative values indicating an early bias, and positive values indicating a late bias. Errors in peak SWE and snowmelt rates are expressed as relative percent error. We use these variables as indicators because they quantify important features relevant to our understanding of snow hydrology and because they provide a more comprehensive analysis of SWE time series performance than the typically used root-mean-square-error and the bias of the entire SWE time series.



**Table 2.** Parameter values used for the three sets of temperature-index model simulations.

| Parameter set | Snowfall threshold (°C) | Snowmelt threshold (°C) | Melt factor (mm °C$^{-1}$ day$^{-1}$) |
|---|---|---|---|
| Single common parameter set for all stations (run 1) | 0.5 | 0 | 3.64 |
| Empirically derived parameter set per station (run 2) | 80th percentile of snowfall temperature observations | 0 | Median melt factor |
| Estimated parameter set per station (run 3) | Equation 5 | 0 | Equation 6 |

## 4 Results

### 4.1 Derived and estimated temperature index model parameters

#### 4.1.1 Snowfall temperature threshold

We find that 89% of all days with snow accumulation, i.e. with $\frac{dSWE}{dt}$>0, occur with mean temperatures at or below freezing (0°C), while only 11% of snow accumulation days (empirically derived from the SWE time series) occur with mean temperatures above freezing (Fig. 3a). There are only 1% of snow accumulation days at or above 5°C. These do not necessarily suggest errors in the data since snow accumulation may occur on days with positive mean temperatures if those days have a strong daily temperature cycle (inducing subdaily freezing conditions) or if humidity is very low, i.e. low wet bulb temperature (Wang et al., 2019), or some combination of both.

If we filter out days with small snow accumulations ($A_t$< 5 mm d$^{-1}$), the mean temperature conditions for snowfall contracts considerably, with only 5% of accumulation days occurring at temperatures below -15°C. This is supported by Fig. 3b, which shows that the magnitude of snow accumulation is limited at temperatures below -15°C, physically consistent with a considerably diminished atmospheric moisture-carrying capacity at such low temperatures. The days with the largest snow accumulation occur between -7°C and -2°C, and accumulation amounts may still be large at $T_t$>+1°C, but quickly diminish for temperatures above this.

In Fig. 3c, we evaluate the quality of snowfall and rainfall days detection at a range of temperatures by comparing daily precipitation amounts observed at the climate stations to the daily changes in the SWE time series from the NH-SWE dataset. A summary of this detailed comparison is provided in Table 3. The results highlight a spectrum in the consistency between snowfall estimated from snow depth observations (NH-SWE) and a simple temperature threshold together with precipitation observations. Below 0°C, only roughly 35% of the days with recorded precipitation ($P_t > 0$) show a corresponding increase in SWE, which is a priori surprising: we would expect most precipitation to fall as snow below freezing air temperatures and accumulate on the ground. However, around 55% of days with $P_t > 0$ and $T_t < 0°C$ do not show an increase in SWE. Figure 3d suggests that days with $P_t > 0$, $T_t < -3°C$ and $dSWE/dt \leq 0$ are days with very small amounts of precipitation ($P_t < 3mm$). On those days, the resulting actual observed snow depth change is likely negligible; given that the SWE time





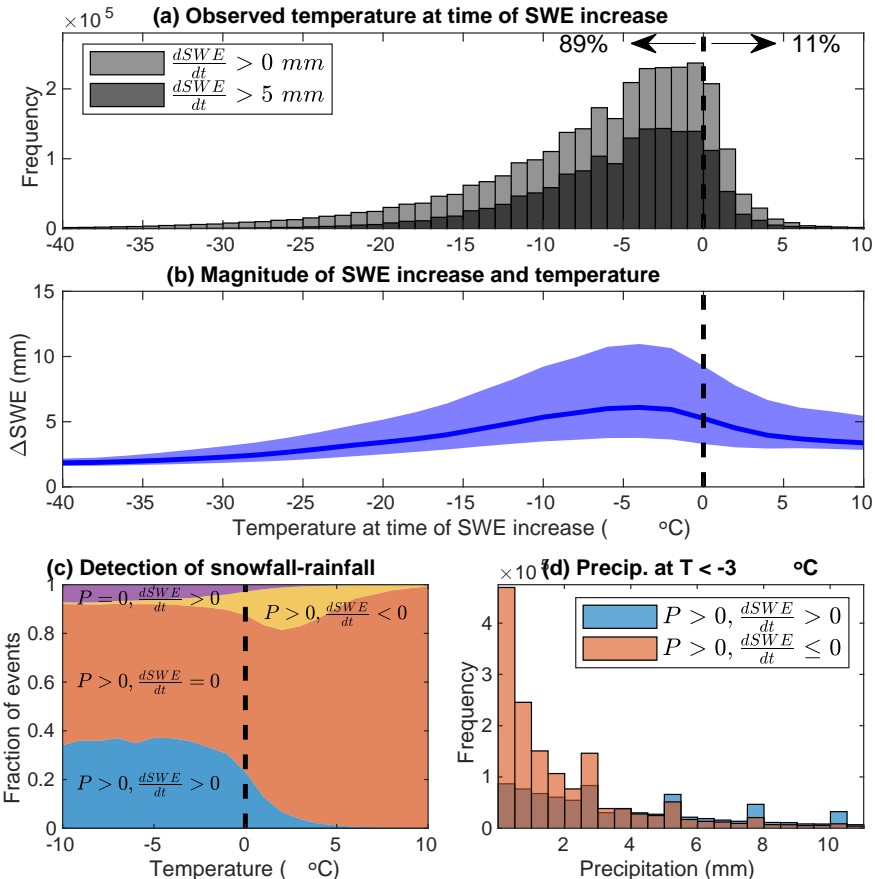

**Figure 3.** Analysis of observed temperatures at the time of snow accumulation. (a) Distribution of daily temperatures at the time of snow accumulation in the NH-SWE time series, for all snow accumulation time steps and for accumulations higher than 5 mm/d only. (b) The magnitude of snow accumulation and daily temperatures. The solid line shows the median and the shaded area the interquartile range for each $2°$C bin between $-40°$C and $+10°$C. (c) Quality of snowfall / rainfall detection as a function of temperature, measured in terms of all possible situations that hint towards snowfall (i.e. precipitation $P > 0$ or $d$SWE/$dt$>0) or rainfall ($P > 0$ and $d$SWE/$dt$=0), including all contradictory events such as $P = 0$ and $d$SWE/$dt$>0). For each $1°$C bin between $-10°$C and $+10°$C, each case is coloured based on its fraction of occurrence. (d) Distribution of snowfall intensities for precipitation events at a temperature below $-3°$C, detected (blue) and not detected (orange) by the NH-SWE time series.

series that we use here (Fontrodona-Bach et al., 2023b) are derived from snow depth observations (and not from actual SWE
220 observations), SWE changes on these days cannot be detected.

In contrast, days with $P_t > 0$ and $T_t$<-3$°$C but $dSWE/dt > 0$ (i.e. days with SWE accumulation) are more uniformly distributed (Fig. 3d). Furthermore, below $0°$C, only about 10% of the days show $dSWE/dt > 0$ and $P_t = 0$, suggesting snowfall





**Table 3.** Summary and description of causes of (un)detected snowfall and rainfall days shown in Fig. 3c. Note that P refers to observed precipitation observed from the climate station time series, while the $dSWE/dt$ refers to changes in the SWE time series from the NH-SWE dataset. Colours in case column refer to Fig. 3c

| Case | Meaning | Suggested cause at $T < 0°C$ | Suggested cause at $T > 0°C$ |
|---|---|---|---|
| P = 0, $\frac{dSWE}{dt} > 0$ (purple) | No recorded precipitation, but an increase in SWE | Snowfall undercatch | Rainfall/Snowfall undercatch |
| P > 0, $\frac{dSWE}{dt} < 0$ (yellow) | Precipitation recorded, but a decrease in SWE | Rain on snow or inaccurate SWE time series | Rain on snow and/or snowmelt |
| P > 0, $\frac{dSWE}{dt} = 0$ (orange) | Precipitation recorded, but no change in SWE | Insufficient precipitation to record an increase in snow depth (Fig. 3c). | Rainfall |
| P > 0, $\frac{dSWE}{dt} > 0$ (blue) | Precipitation recorded, and an increase in SWE | Snowfall | Snowfall and Rainfall |

undercatch by the climate station, something widely reported in the literature (Adam and Lettenmaier, 2003; Kochendorfer et al., 2020; Pan et al., 2020).

225  Above $0°C$, most days with $P_t > 0$ show $dSWE/dt = 0$, which is expected if precipitation is dominated by rainfall. It is also noteworthy that 20% of days with $1°C < T_t < 4°C$ show $P_t > 0$ and $dSWE/dt < 0$: this relatively large fraction represents days with rain-on-snow conditions, which could potentially enhance snowmelt rates and snowmelt days (Cohen et al., 2015).

  The snowfall temperature threshold (empirically derived from SWE time series and corresponding temperature observations) varies across climates (Fig. 4). Very cold climates (high $\Delta_T$ and low $\overline{T}$) have most snow accumulation days at temperatures

230 below $0°C$, but more temperate climates have a larger fraction of snow accumulation days at $0°C < T_t < 5°C$. While increasing the temperature threshold for snowfall might capture more of those above-freezing snowfall days, it also increases the false positive rate, i.e. recording actual liquid precipitation as snowfall. Estimating the threshold temperature $T_a$ for snowfall from climate variables and a multilinear regression model via Ordinary Least Squares (see Section 3.3) yields a coefficient of determination of $R^2 = 0.67$ and the following coefficients:

235 $T_a = 0.199 \times \overline{T} - 0.361 \times \Delta_T + 2.512$               (5)

where $\overline{T}$ is the mean annual temperature of a station, and $\Delta_T$ is the amplitude of the annual temperature cycle at a station (Table 1). However, we set the minimum threshold value to $0°C$, as the model estimates snowfall thresholds below $0°C$ for the colder climates. The estimated values are used in one of the three temperature-index model simulations of our simulation workflow (Fig. 2).





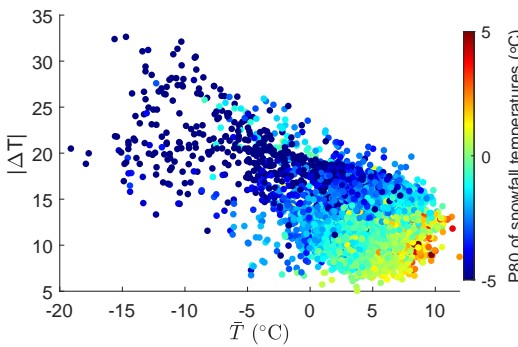

**Figure 4.** Snowfall temperature thresholds and climate variables (one point per station). $\Delta_T$ is the temperature amplitude of the climate and $\overline{T}$ is the mean temperature of the climate (see section 3.1). Colours show the snowfall threshold, defined as the 80th percentile of the observed daily temperatures on days when there is snow accumulation at each of the sites (see Section 3.3).

### 4.1.2 Snowmelt temperature threshold

Figure 5a shows that 76% of daily SWE decreases occur at daily temperatures above 0°C and 24% at temperatures below 0°C. While snowmelt is only be expected above freezing conditions, it is not unreasonable for snowmelt to occur at temperatures close to but below 0°C, as the mean daily temperature may not capture subdaily positive temperature excursions or days near freezing conditions with high incidence of shortwave radiation. However, days with SWE decreases during days with negative temperatures might also indicate other processes like sublimation, snow redistribution, or inaccuracies in either the temperature data or the SWE time series.

Above 0°C, the magnitude of SWE decreases scales linearly with temperature (Fig. 5b). Below 0°C, SWE decreases are low in magnitude and their frequency is small compared to the frequency of days with SWE increases or no change in SWE (Fig. 5c). It is interesting to note that the distribution of the occurrence of daily SWE increases and of SWE decreases cross each other around 0°C (blue and read lines in Fig. 5c), suggesting that below 0°C it is more likely to have snow accumulation or no change in SWE, and above 0°C it is more likely to have snowmelt days.

### 4.1.3 Melt factor

Daily melt factors show a wide, but positively skewed, distribution of possible values and with a peak between 2 and 4 mm °C$^{-1}$ day$^{-1}$ (Fig. 6a). The median of all annual melt factors (2.93 mm °C$^{-1}$ day$^{-1}$) is slightly lower than for the melt season only (3.64 mm °C$^{-1}$ day$^{-1}$). This is likely because SWE decrease during the accumulation season typically occurs at lower net radiation inputs, and therefore melt factors (and melt rates) are lower.

Melt factors vary with mean melt season temperatures (Fig. 6b,c). The range of possible melt factors is very large at mean melt season temperatures close to 0°C, and most melt seasons have mean temperatures between 0°C and 5°C (Fig. 6b). The melt factor range converges to 3 to 5 mm °C$^{-1}$ day$^{-1}$ with increasing melt season temperatures (Fig. 6b). Only 17% of stations





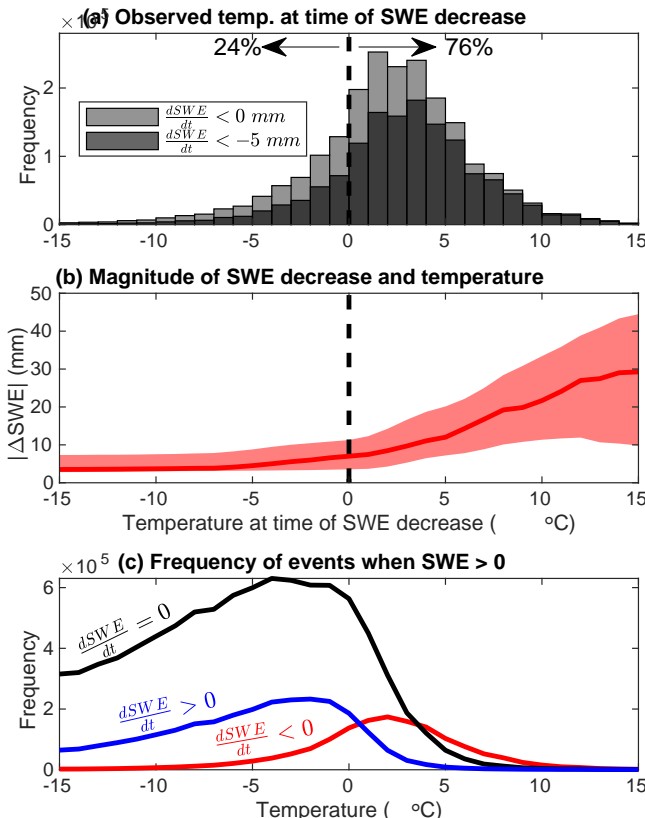

**Figure 5.** Analysis of observed temperatures for days with SWE decreases. (a) Distribution of daily temperatures for days with SWE decrease in the NH-SWE dataset, either for all days with $dSWE/dt$<0mm or for days with $dSWE/dt$<-5mm. (b) Magnitude of SWE decrease and daily temperatures. The solid line shows the median and the hatching shows the interquartile range, for each 1°C bin between -15°C and +15°C. (c) Distribution of daily SWE changes ($dSWE/dt$<0,$dSWE/dt$>0 or $dSWE/dt$=0) as a function of temperature. Lines based on histogram bins every 1°C bin between -15°C and +15°C.

have mean melt season temperatures below freezing (Fig. 6c). These stations with cold melt seasons show higher melt factors and a high interannual variability, limiting the predictive capacity of temperature-index models for these conditions. In contrast, 83% of stations have positive mean melt season temperatures and lower melt factors, converging to 3 and 5 mm °C$^{-1}$ day$^{-1}$ and with far lower interannual variability.

We explore the variability across stations and across years in the mean melt season melt factor alongside climate variables
in Fig. 7. The range of possible melt factors is very wide, from 1 to 12 mm °C$^{-1}$ day$^{-1}$, for stations with annual snowfall below 300 mm but converges towards 3-5 mm °C$^{-1}$ day$^{-1}$ for stations with annual snowfall above 300 mm (Fig. 7a). This convergence is also observed when comparing the melt factor and elevation (Fig. B3). Figure 7b shows that the mean melt





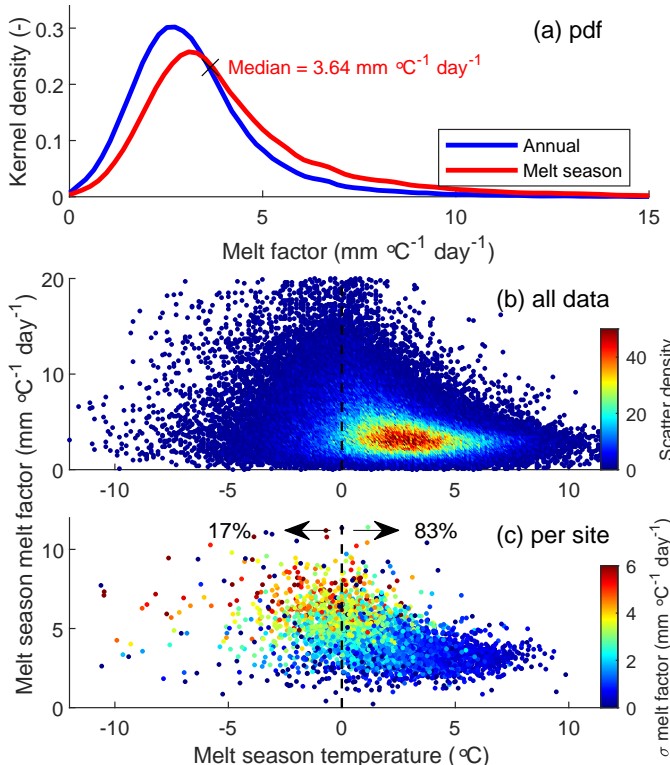

**Figure 6.** Empirically derived melt factors and the link to temperature and interannual variability. (a) Distribution of observed annual and seasonal melt factors. (b) Median melt season melt factor and mean melt season temperatures (all station-years). (c) Per station, mean melt factor, mean melt season temperature, and interannual variability ($\sigma$) of the melt factor.

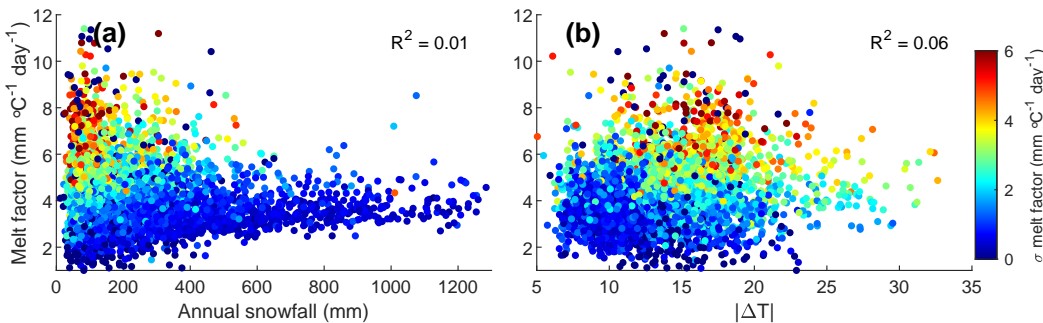

**Figure 7.** Climatic variability of the mean melt season melt factor. (a) Melt season melt factor and mean annual snowfall for each station; (b) Melt season melt factor and yearly amplitude of the temperature cycle. Both a and b are coloured by the interannual variability ($\sigma$) of the melt season melt factor.





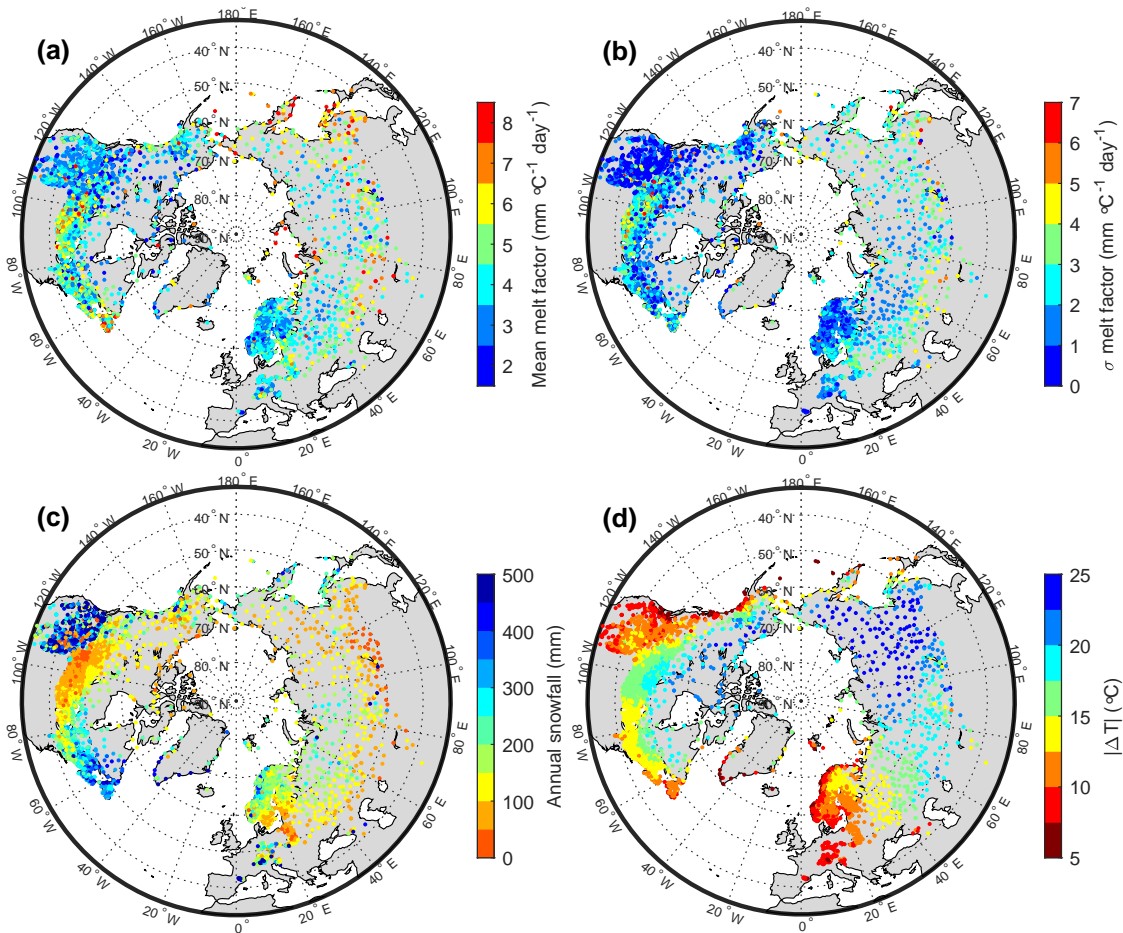

**Figure 8.** Spatial distribution of the melt factor (a), its interannual variability ($\sigma$) (b), annual snowfall (c), and the amplitude of temperature $\Delta_T$ (d).

factor and its interannual variability increase weakly with $\Delta_T$; this is mostly driven by colder, more continental climates, with low precipitation, shallow snow, and late melt onset having higher melt factor values.

Spatially, the convergence of melt factor values to 3-5 mm $°\mathrm{C}^{-1}$ day$^{-1}$ and low interannual variability (Fig. 8a,b) clearly occurs over the areas with higher annual snowfall (Fig. 8c), and the more maritime and temperate climates (Fig. 8d), namely the mountain areas of Western North America, the European Alps, the Pyrenees and Scandinavia. Outside these deeper snowpack areas, over colder climates with shallower snowpacks, such as continental North America and Eurasia, it is clear that melt factors are higher and spatially more variable.

Given the variability of the melt factor with melt season temperatures (Fig. 6), annual snowfall (Fig. 7) and its geographical distribution (Fig. 8), it was interesting to check whether a multilinear regression model can be used to estimate the melt factor based on climate and geographic variables. Among the variables tested ($\overline{T}$, $\Delta_T$, mean annual snowfall, latitude and elevation)





the best model shows, however, a very low coefficient of determination, $R^2$ = 0.16, suggesting such an approach offers very low predictive skill. The model equation reads as follows:

$$\alpha = 9.6 - 0.00083 \times e - 0.0868 \times L - 0.117 \times \overline{T} \qquad (6)$$

where $\alpha$ is the melt factor [$°C^{-1}$ day$^{-1}$], $e$ [m] is the elevation above sea level, $L$ [decimal degrees] is the latitude, and $\overline{T}$ [°C] is the mean annual temperature. Our melt factor estimation model shows that melt factors decrease with elevation, which is opposite to what Hock (2003) suggested. This probably results from the convergence towards lower melt factor values (3-5 $°C^{-1}$ day$^{-1}$) that our analysis shows for deeper snowpacks, which are common at higher elevations. Similarly, the melt factor

also decreases with increasing latitude. The estimated melt factor increases with lower mean temperatures as the shortwave radiation component for melt becomes more important. We use these estimated melt factors, together with the estimated snowfall thresholds from Equation 5 for the third set of temperature-index model simulations (see Fig. 2).

### 4.2  Temperature-index model performance

The performance of the temperature-index model is generally very good (Fig. 9), especially considering the simplicity of the

model and that the parameters were empirically derived from the data or estimated from climate variables but not calibrated to improve performance. Importantly, all snow season timing variables show median errors close to zero days, and interquartile ranges do not exceed a 15-day error (Fig. 9a,b,c). Especially excellent is the performance of the timing of the accumulation season onset, where even the whiskers do not exceed a 15-day error for any of the three sets of model simulations. Regarding the snowmelt season onset (or timing of Peak SWE), all three simulations show a median timing that is 4 days too early.

However, it must be noted that the validation data (the NH-SWE dataset) showed an average model delay of 2 days in the onset of the melt season (Fontrodona-Bach et al., 2023b), which suggests that the median error in Fig. 9b may actually be closer to zero. Given that the snowmelt season onset depends mostly on the snowmelt threshold temperature, which is the same for all simulations, it is unsurprising that all three simulations show an almost equal performance for this variable. For the end of the snow season timing, the differences between the simulations are more visible but still relatively minor. This minor difference

is due to the high dependence on the melt factor, which is estimated differently in all three sets of simulations.

Simulations with the common parameter set have a better median error, likely because the melt factor used is the median of all stations. Simulations with the empirically derived parameter set have a narrower interquartile range, as values are empirically derived from observations at each station individually. Simulations with the estimated parameter set have a slightly worse, though comparable, performance.

Regarding snow season magnitude variables, the temperature-index model shows good median errors across variables and parameter sets, but also has large interquartile ranges and whiskers, and many outliers (Fig. 9d,e,f). Interquartile ranges for peak SWE are within 45% error, and the median values display a negative bias at around -10% error. No significant differences arise between parameters sets for peak SWE. For snowmelt days, simulations using the empirically derived and estimated parameter sets have a similar median of 10% and 8% error, while simulations using the common parameter set performs slightly worse

with a median 20% error overestimation of the number of snowmelt days. For snowmelt rate, the common parameter set also





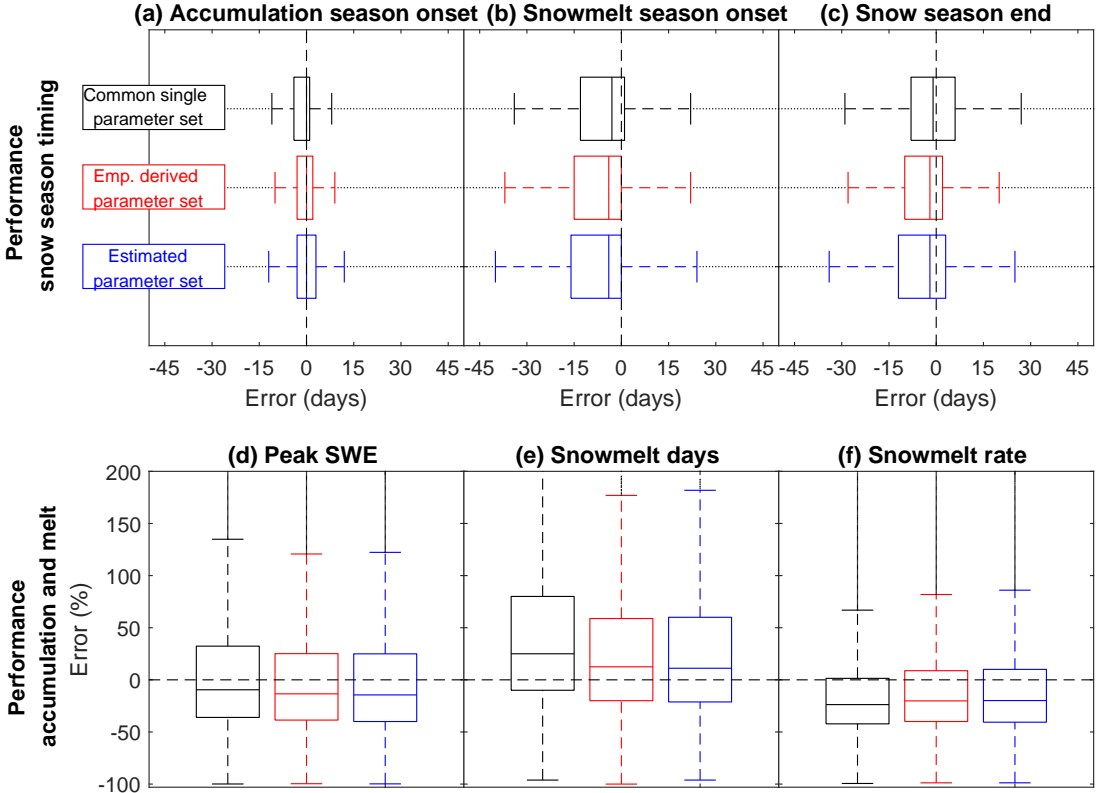

**Figure 9.** Temperature index model performance. The three sets of model simulations are shown in each boxplot, namely the common parameter set for all stations, the empirically derived parameter set for each station, and the estimated parameter set for each station. Boxes are built with all station-years in the analysis. Top row (a,b,c): performance of the three snow variables related to timing of the snow season. Bottom row (d,e,f): performance of the three snow variables related to magnitudes of the snow season. Note that outliers are shown as points outside the box whiskers.

performs slightly worse (-25% error) than the empirically derived and estimated ones (-22% error). Simulations across all parameter sets underestimate the snowmelt rate but have the narrowest interquartile range of the three snow season magnitude variables.

Given the overall similar performance between the three sets of simulations, only peformance results using the predicted
parameter set are shown in Fig. 10. This demonstrates clear spatial differences, where snow season timing variables (Fig. 10a,c,e), show the mountain areas of Western North America as well as Scandinavia have poorer performance, with late accumulation season onset, early melt season onset, and early end of the snow season. Performance is much better over the rest of the Northern Hemisphere and for all three timing variables.

Performances of snow accumulation and melt magnitudes show marked differences between Northern Hemisphere regions,
especially for peak SWE (Fig. 10b). Many stations underestimate peak SWE over the United States and over Europe, while



overestimates are observed over Canada and Eurasia. However, over these stations the underestimation of peak SWE does not inherently lead to fewer snowmelt days, since the snowmelt rate is also underestimated. In contrast, data from stations across the rest of the Northern Hemisphere mostly overestimate the number of melt days (Fig. 10d), suggesting that the melt temperature threshold or melt factor might not be accurate for these stations. Figure B2 suggests melt onset initiation actually

peaks at air temperatures between 1°C and 3°C, which highlights a small but significant disconnect between air and snow temperatures that may also explain the negative bias in the timing of melt onset using the zero degrees threshold assumption. Finally, melt rates show both the smallest and most spatially consistent relative errors of all three variables (Fig. 10f), generally not exceeding 10%. The melt rate errors also have some spatial structure, with underestimation generally occurring over regions with deeper snowpacks, such as Western North America, Scandinavia, and the European Alps, and overestimation over regions

with shallower snowpacks, such as continental Canada and Eurasia.

Overall, the simple temperature-index model performs remarkably well for most input data and stations, but there can also be large variance, and it occasionally performs poorly. Examples of this range of excellent, good, acceptable and poor performances are provided as time series in Fig. 11. In the next section, we discuss potential reasons for the range of performances.





**Figure 10.** Spatial distribution of temperature-index model performance. Left column (a,c,e) for snow season timing variables, and right column (b,d,f) for snow season magnitude variables. The median error for each station is shown for the simulation using the estimated parameter set.



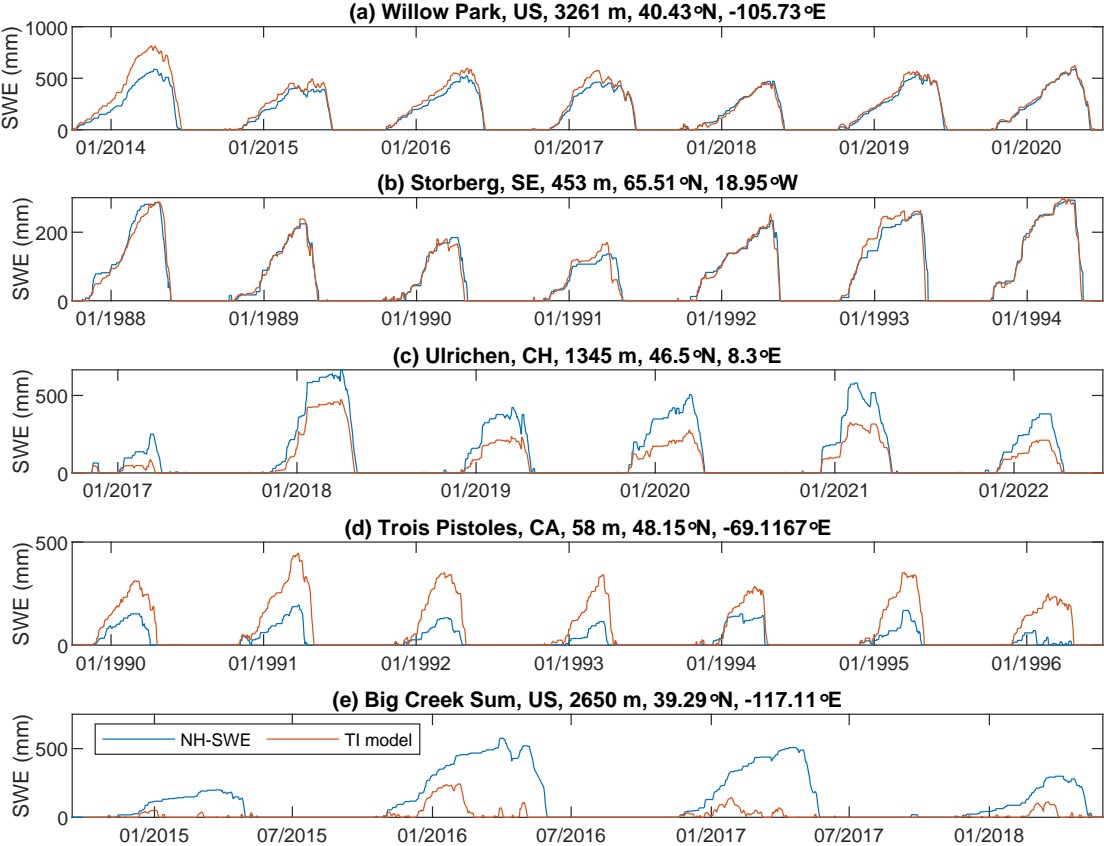

**Figure 11.** Example SWE time series comparisons of NH-SWE vs temperature-index model simulations with the estimated parameter set. Good (a,b), medium (c) and poor (d,e) performances across different regions. Title of each subpanel shows the station ID, elevation in metres, latitude and longitude.





## 5   Discussion

### 5.1   Snowfall temperature thresholds

Our analysis highlights the range of temperatures over which snowfall occurs and shows that most snowfall days occur at daily average temperatures below freezing. This is broadly in accordance with previous studies (Rohrer and Braun, 1994; Magnusson et al., 2019), but we now provide this evidence at much larger spatial and temporal extents. This highlights the potential of a simple snowfall air temperature threshold to capture most snowfall days successfully, especially for cold, continental climates with shallow snowpacks (e.g., Eurasia, continental Canada). However, our results also highlight that some relatively large snow accumulation days at daily average temperatures close to, but just above freezing temperatures, are difficult to capture. The risk of missing snow accumulation days can be higher over warmer maritime climates with deeper snowpacks (e.g. western North American mountains, the European Alps, or Scandinavia) at near freezing temperatures. Therefore, where snowfall days above freezing temperatures are critical to capture, we emphasise the importance of using a different precipitation phase method, such as the use of wet bulb temperature (Jennings and Molotch, 2019), if available. Alternatively, snowfall temperature thresholds may be adjusted regionally, as our analysis in Figures 3 and 4 shows that warmer climates generally have higher snowfall temperature thresholds.

While a temperature-based precipitation phase threshold may miss snowfall days above freezing, they are well captured by snow observations. Conversely, snowfall days are well captured at temperatures well below the snowfall threshold. It is important to note that time series of snowpack observations may also fail to register precipitation days if they are small (< 3 mm), which is typical of cold climates, such as Eurasia and continental Canada. Additionally, our analysis suggests that rain-on-snow days during the melt season are more frequent and widespread than previously recognised. Undercatch of solid precipitation in rain gauges is also observed (ca. 10% of fraction of precipitation days below freezing).

### 5.2   Melt temperature thresholds and melt factors

We have tested the validity of a simple temperature threshold of 0°C to capture snowmelt using an observation dataset at the largest spatial and temporal scale to date. The data clearly indicate that snowmelt occurs almost exclusively at daily average air temperatures above 0°C, as evidenced by the sharp decrease in the frequency of days with no change in SWE as temperatures rise above this value and the linear increase of snowmelt magnitude with temperature (Fig. 5). Overall, our results highlight snowfall and snowmelt days reach an equal frequency of occurrence at air temperatures around 0°C, suggesting this is an appropriate air temperature threshold for most data in a large-scale study. However, we also find that to accurately predict the onset of the snowmelt season, a slightly higher air temperature threshold may be required, as our analysis in Fig. 9 shows snowmelt season onset is estimated on average 4 days too early. This is likely because zero degree snowpacks still need to overcome the latent heat of fusion to initiate snowmelt (Molotch et al., 2009; Jennings et al., 2018a), and thus air temperatures may be slightly higher than zero during actual melt initiation. However, temperature-index snow models often choose a value of 1°C (Avanzi et al., 2022), 0°C (Senese et al., 2014) or even -1°C (Elias Chereque et al., 2024) as the threshold for snowmelt initiation. It must also be noted that the NH-SWE time series used for validation exhibited a slight delay on average in snowmelt





season onset (Fontrodona-Bach et al., 2023b), thus it is possible to speculate this could reduce the overall uncertainty in snowmelt onset for the temperature-index model results presented here.

We have shown that melt factors vary substantially across spatial and temporal scales. Seasonal scale melt factors across the Northern Hemisphere ranged from 1 to 12 mm $°C^{-1}$ day$^{-1}$, in accordance with ranges estimated in the literature from individual sites at smaller spatial scales (Braithwaite, 1995; Kane et al., 1997; Lefebre et al., 2002; Hock, 2003; Braithwaite, 2008; Shea et al., 2009; Asaoka and Kominami, 2013). Within this range, there is a clear dominance of melt factors that occur between 3 and 5 mm $°C^{-1}$ day$^{-1}$, particularly in regions with deeper snowpacks and more temperate climates. This key finding suggests that melt factors tend to stabilise within these specific climatic conditions, which also tend to have warmer 375 melt seasons. Furthermore, the low interannual variability of these melt factor values suggests that climatic conditions with deep snowpacks and temperate climates are conditions that are well suited for temperature-index modelling. In contrast, it may be harder to robustly estimate melt factors for colder, continental climates with shallower snowpacks, as they typically exhibit larger melt factors with higher interannual variability. In addition, our results suggest other loss mechanisms, such as sublimation or snow redistribution by wind, are more common over these colder, shallower snowpack climates. As a result, 380 temperature-index melt modelling will likely provide a poorer representation of SWE dynamics in these conditions, as other accumulation or loss mechanisms that are typically more important during the accumulation season may not be captured by the temperature-index approach.

### 5.3 Long-term, hemispheric scale temperature index model performance

The median performance of the temperature-index model across various indicator variables for a wide range of climate condi-
tions across the Northern Hemisphere is surprisingly good given the model's simplicity and that parameters were only estimated from data, and not calibrated to improve model performance. Evaluation of snow season timing variables across the Northern Hemisphere showed a median error of 0 days for the snow season onset, 4 days early for the snowmelt onset, and 1-2 days early for the end of the snow season. This shows the long-term time series dynamics are, on average, well captured by the temperature-index approach. Regarding snow magnitude variables, a median relative error of -10% was found for peak SWE,
a median relative error of +10% for the number of snowmelt days, and a median relative error of -22% for the snowmelt rate. These results highlight the excellent performance that the temperature-index models can achieve if the input data is of high quality and if most snowfall days occur at temperatures below 0°C.

The temperature-index model, on average, underestimates peak SWE, especially over deeper snow climates (e.g. western North American mountains, the European Alps, and Scandinavia), where a negative bias is observed for up to 75% of stations.
This is consistent with our analysis and highlights a possible combination of undercatch of solid precipitation, the inability of the temperature threshold to capture snowfall days above freezing, the potential underestimation of the snowmelt initiation temperature threshold, climate station measurement errors, or residual uncertainties with the validation dataset. Nevertheless, it is challenging to point out the exact cause of smaller errors on peak SWE. Furthermore, feedbacks to other metrics arising from errors in peak SWE may occur. For instance, a temperature-index model underestimation of peak SWE may, in turn,





result in an underestimation of the number of melt days, but an underestimation of the snowmelt threshold temperature may overestimate the number of days with melting conditions and compensate for the underestimation of melt days.

Lastly, the temperature-index model performance analysis suggests that the estimated melt factors are relatively robust. Snowmelt rates, which is the model performance indicator most sensitive to melt factor accuracy, have smaller relative errors than peak SWE and the number of melt days. However, it may also suggest that despite the challenges in estimating melt factors

for colder and shallower snowpacks (as outlined above), melt rate model performance is not overly sensitive to melt factor accuracy, as melt rate estimates are consistently robust even under conditions where the melt factor experiences high interannual variability. In any case, an important implication is that melt rates can be more confidently estimated using temperature-index models than some of the other key snow metrics (such as peak SWE and snowmelt onset) evaluated here. However, it is important to note that any trend introduced by climate change impacts may complicate the robust estimation of melt factors

(Raleigh and Clark, 2014; Musselman et al., 2017).



## 6    Conclusions

This study comprehensively examines the parameters of the temperature-index snow model, their performance, and their limitations across diverse climatic conditions. The findings highlight the significance of temperature thresholds in governing snowfall and snowmelt events and their simulation using a simple temperature-index model. While a temperature-based precipitation

phase threshold captures most snowfall events on a daily scale, the risk of missing substantial snowfall events, particularly in climates receiving precipitation near freezing temperatures, must be considered. The assimilation of snow data, such as the NH-SWE dataset, into temperature-index modelling efforts emerges as a future line of research, owing to its ability to capture snowfall events above freezing temperatures or rain-on-snow events.

Evidence is clear that most snowmelt occurs at daily temperatures above freezing and that SWE decreases below freezing in

the NH-SWE dataset are likely not snowmelt but sublimation or snow redistribution processes. The study highlights the equal frequency of snowfall and snowmelt events occurring at a temperature of 0°C, making this a suitable threshold for large-scale studies. However, our results also suggest the potential need for a slightly higher temperature threshold to initiate the snowmelt season.

The investigation of melt factors offers insights into their variability across climatic conditions. Melt factors converge to

3-5 mm °C$^{-1}$ day$^{-1}$ in regions with deeper snowpacks and temperate climates. Our study highlights challenges in estimating melt factors for colder, continental climates with shallower snowpacks due to higher interannual variabilities. Still, the model performance was not worse over those regions.

The temperature-index model consistently captures snow season timing variables, although minor deviations in melt season initiation and end are noted. Our results highlight the model's challenges in estimating peak SWE and the number of melt days,

particularly in regions with deeper snow climates, attributed to factors including undercatch of solid precipitation, temperature thresholds, and potential data errors. However, the robustness of estimated melt factors yields accurate melt rate predictions.

In summary, this study provides valuable insights into temperature-index modelling for applications across diverse snow climates. Our results should help refine modelling approaches by acknowledging the interplay of temperature thresholds, snow cover dynamics, and the robust melt factor estimations. Further research could advance our understanding of snowpack

responses to global warming by applying the temperature-index melt model on a global scale, using globally available climate station data, and assessing the sensitivity of various snow climates to warming scenarios.

*Code and data availability.* The daily temperature and precipitation time series used in this study are available from: the Global Historical Climatology Network daily (GHCNd) at https://www.ncei.noaa.gov/pub/data/ghcn/daily/ (Menne et al., 2012); the European Climate Asssessment and Dataset (ECA&D) at thttps://www.ecad.eu/dailydata (Klein Tank et al., 2002); the Meteoswiss portal IDAWEB at

https://gate.meteoswiss.ch/idaweb (MeteoSwiss, 2023); the Bias Corrected and Quality Controlled SNOTEL (BCQC SNOTEL) at https://www.pnnl.gov/data-products (Yan et al., 2018; Sun et al., 2019). The NH-SWE daily snow water equivalent time series used in this study are available at https://zenodo.org/record/7515603 (Fontrodona-Bach et al., 2023a, b). The code used to run the model simulations and generate all figures in the manuscript is available at https://github.com/Fontro/TI_modelling.git.





**Appendix A: Gap filling and quality control**

We apply a simple gap-filling and quality control procedure for the temperature time series, similar to Serreze et al. (1999). For each station, an average daily climatology is calculated, and we remove values that are more or less than three times the standard deviation of the daily temperature climatology, which is a conservative estimate. We also remove temperatures above or below +/- 55 °C. We fill gaps of one day by linear interpolation, and gaps of two days with the corresponding neighbour. For larger gap sizes, we fill the gap with the climatological mean, but only if the length of the gap is less than 20% of the

climatological winter. The climatological winter is defined as the period with temperatures lower than the mean temperature minus two times the standard deviation of daily temperatures. We do not gap-fill the precipitation time series, but we remove daily precipitation values higher than 176 mm, considered outliers. This value corresponds to the 99.99th percentile of all daily precipitation observations over all gathered precipitation time series.

**Appendix B**

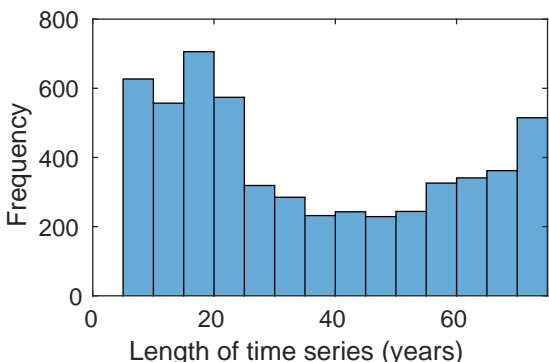

**Figure B1.** Length of station time series in this study.



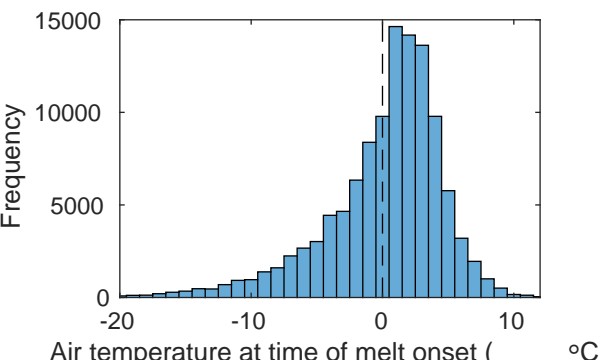

**Figure B2.** Distribution of daily temperatures at the time of snowmelt season onset. Median is +1.10°C

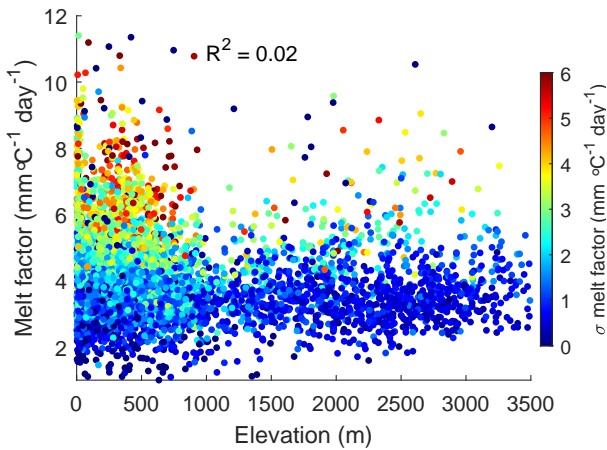

**Figure B3.** Median melt factor and elevation, coloured by its interannual variability ($\sigma$).



*Author contributions.* AFB collected and processed data, developed the model parameterisations and run model simulations, produced all the figures and results, and wrote the manuscript. JL and BS provided an initial data collection and model code. JL closely supervised the underlying PhD research and wrote the initial PhD project proposal. JL, BS and RW contributed to editing the text and gave regular input on the work and the manuscript process.

*Competing interests.* The authors declare that they have no conflict of interest.

*Acknowledgements.* AFB acknowledges funding from the UK's Natural Environment Research Council (NERC) CENTA2 doctoral training program, grant number NE/S007350/1. AFB acknowledges support from the School of Geography, Earth and Environmental Science research fund. The computations described in this paper were performed using the University of Birmingham's BlueBEAR HPC service, which provides a High Performance Computing service to the University's research community. See http://www.birmingham.ac.uk/bear for more details.



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
