# Peer review of "Estimating robust melt factors and temperature thresholds for snow modelling across the Northern Hemisphere"

_EGUsphere, 2025_

## Author Comment (AC1)

In black: Referee comments
In blue: Authors' response

**RC1: 'Comment on egusphere-2025-1214', Anonymous Referee #1, 24 Mar 2025**

The temperature index method is a convenient and widely used method in snow simulation. This study estimates the important parameters in the temperature index method based on the published SWE and climate datasets, and analyzes the influence factors of these parameters. The results can provide insights into the temperature index method, making this paper worth publishing in HESS. Having said that, I would like to point out some major concerns that should be addressed before publication.

Dear referee, we thank you for your fast, thorough and constructive review of our paper. We appreciate you finding the paper worth publishing in HESS, and **we will gladly address the concerns** you have pointed out to make the study better. See below detailed responses to your comments.

1. Potential Circular Logic: The temperature threshold and melt factors are estimated based on the SWE dataset, which is subsequently used to evaluate the performance of the temperature index model. This approach risks circular reasoning. A more rigorous method would involve dividing the dataset into two subsets—one for parameter estimation and another for model validation.

We agree that some circular logic could be interpreted here, as the SWE dataset is used both for parameter estimation and for evaluation. As Sections 3.3 and 3.4 describe, there are three different parameter sets and model simulations for each station in the study (illustrated in Figure 2 and Table 2 in the manuscript). We respond to your point separately for each parameter set.

**The first set** simulates SWE with the same parameter values for all stations across the Northern Hemisphere. The aim of this is to evaluate how one single parameter set performs when applied to all stations across the Northern Hemisphere. It is true that the melt factor value used (3.64 mm/°C/day) is the median of all empirically derived melt factors. We have randomly subsampled ½ of the dataset 1000 times and the computed median melt factors ranged from 3.61 to 3.66. Therefore, we doubt the results would change if we did that for this first set of simulations.

The aim of **the second set** of model simulations is to evaluate the model performance using empirically derived parameter values for each station individually. In other words, we evaluate the best possible parameter set, based on observations. Here, splitting the dataset spatially is not possible as a parameter needs to be estimated for each station. Instead, **we will split each station temporally** and use half the data for empirically deriving the parameter, and the other half for model performance evaluation.

In **the third set** of model simulations, we use estimated parameter values from multilinear regression models based on climate variables. Here, we agree that we should spatially split the dataset in two subsets. The first subset (⅔ of the available stations) will be used to build the regression model that estimates model parameters based on climate variables,

and the second subset (⅓ of the available stations) will be used for model evaluation. **We will modify this and include the new results in the revised manuscript.**

2. Subjectivity in Determining the Melt Threshold: In Section 4.1.2, the melt threshold appears to be assumed as 0°C, with supporting analyses provided. However, if the threshold were slightly adjusted around 0°C, the conclusions in Section 4.1.2 would still hold. To minimize subjectivity, a quantitative approach should be employed. While the melting process is expected to occur at 0°C from a physical standpoint, the temperature data used do not precisely reflect the conditions at the exact location where phase changes occur (e.g., the snow surface for melting and the atmosphere for precipitation partitioning).

We agree that there is subjectivity in determining the melt threshold. In the literature, this threshold varies between -1°C and 1°C, so we heuristically chose 0°C as the central value of this range, and we supported it with our analyses in Section 4.1.2 (Figure 5 in the manuscript). However, in this section a quantitative approach is not plausible due to data limitations. This limitation is discussed in lines 161-166 (Section 3.3), and owes to the fact that the NH-SWE time series cannot confidently distinguish if a SWE decrease (at the daily time scale) is due to snowmelt, snow redistribution, sublimation, or a data error, and therefore limits the capacity for this more quantitative approach.

Nevertheless, the point of the reviewer remains extremely valid, and our analyses indeed support that a mean daily temperature of ~1.1°C might be more appropriate for accurately estimating the onset of the snowmelt season (see Figure B2), as this may better reflect the snowpack becoming isothermal. On the other hand, a varying melt threshold temperature will add a free parameter to our approach and add complexity to our broader aim to provide robust estimates for melt factors across the Northern Hemisphere. **We will analyse the sensitivity of model performance to a varying melt threshold temperature and we will include it and discuss it in the revised manuscript.**

3. Clarity in Time Scale: The study computes temperature thresholds and melt factors at the daily scale in some instances, while averaging them in others. This inconsistency makes the methodology difficult to follow. A clearer distinction between different time scales should be provided.

You are right that the time scales across sections and figures might be confusing. The analyses of the two melt thresholds and the melt factor start at the daily time scale by comparing the daily time series of temperature and precipitation with the daily time series of SWE. This leads to the insights on temperature thresholds and melt factor at the daily time scale from Figure 3, Figure 5, and Figure 6a. However, in this study we use temperature-index modelling with constant parameters (non-varying in time, only in space), therefore we compute the median of the daily temperature thresholds and melt factor to obtain a seasonal value for each year (e.g. Figure 6b). The calculated seasonal values are further averaged to obtain a mean temperature threshold and melt factor for each station (that is Figure 4, Figure 6c, Figure 7, Figure 8). In the revised manuscript, **we will make this clearer in the methods section, as well as in each subsection and each Figure.**

4.  Effectiveness of a Single Parameter Set: The model employing a common single parameter set outperforms the other two models in certain aspects. This raises the question of whether complex parameter estimation methods are necessary. A discussion on the added value of these methods compared to a simpler approach would be beneficial.

We agree that a more thorough discussion about this is needed in the manuscript, and we find it encouraging that a simple parameter set has good performance and provides a useful justification for studies with no or little physical information on which to base their model. On the other hand, we also want to understand and provide some physical basis for the spatial variability of the melt factors and their performance. There are a variety of reasons leading to small differences in model performance between the three parameters sets. First, the melt temperature threshold is the same for all sets of simulations, which limits the possible differences in results. Second, the snowfall temperature threshold varies per station but only minimally, as we set a minimum value of 0°C (for which we have now added a justification, in response to one of your minor issues from L257). For the empirically derived and estimated parameter sets, 79% and 86% of stations have a threshold value of 0°C, respectively. For the rest of the stations, values are mostly below 2°C (see Figure 1 below). The small variability of the snowfall temperature threshold across parameter sets may also contribute to the small differences in model performance across parameter sets. Note that the figure excludes stations with a snowfall threshold of 0°C.

[Figure]

Figure 1. Distribution of the snowfall threshold in the empirically derived and estimated parameter set, excluding stations with a 0°C threshold (which represent 79% and 86% of stations for the empirically derived and estimated parameter set, respectively).

The last factor of model performance variability across parameter sets is the melt factor. This is the parameter that varies most between stations and between parameter sets, as seen in Figure 2 left panel below. The melt factor differs between the two parameter sets (Figure 2 right panel) because our parameter estimation model based on climate variables has low predictive skill (L275 in manuscript).

[Figure]

Figure 2. Melt factor distribution in the empirically derived parameter set vs the estimated parameter set based on climate variables.

The model performance variables most sensitive to the melt factor are the melt rate and the time of the end of the snow season. However, there is a strong correlation in model errors between the two sets of model simulations (see Figure 3 below). This indicates that model performance is not very sensitive to model parameters, at least when evaluated over long-term time series, as we do in this study.

[Figure]

Figure 3. Model errors in the empirically derived parameter set vs the estimate parameter set based on climate variables. Melt rate error (left) and timing of end snow season error (right). Dashed line indicates the 1:1 line.

As you point out, this suggests that for long-term studies using temperature-index modelling, it is not necessary to use complex parameter estimation methods, as the use of commonly applied values in the literature will lead to similar model performances. **We will add this discussion to Section 5.3.**

Minor issues:

Thank you for pointing out these minor issues. We will incorporate your suggestions accordingly.

L106: Provide the full name of SNOTEL. Will do.

L123: Ensure consistency in terminology (e.g., CHCN-d, CHCNd, and CHCN-Daily). Will do.

3.1: Make it clear what you are specifically referring to here. There are more than two indices in the Table 1. We will specify whether each term is a snow season term or a climate index.

L162: "toe" should be "to". Indeed.

L163-166: Difficult to understand. Please rephrase and explain it more clearly. Will do.

3.4: Consider merge sections 3.2 and 3.4, moving the descriptions of the model to the beginning of 3.4 section.The rationale behind this decision was that it was confusing to start describing the estimation of model parameters, without having initially described the model we are using.

Figure 3d: what does the dark orange mean? - This is just the overlap of orange and blue bars. We will add this detail in the caption.

L257: Why is a threshold lower than 0℃ not allowed? - We assume the reviewer means L235? Due to our definition of snowfall threshold in lines 152-156 in the manuscript, our snowfall threshold estimation model predicts threshold temperatures below 0℃. However, based on Figure 3 in the manuscript, precipitation below 0℃ is very likely to be snow. We therefore only allow the snowfall threshold to be 0℃ or higher, to avoid capturing too much snowfall as rain.

In some figures, the text is overlapped. Please check and modify them. Thank you for pointing this out, we will revise the figures.

---

## Author Comment (AC2)

**In black: Referee comments**
**In blue: Authors' response**

**RC3: 'Comment on egusphere-2025-1214', Anonymous Referee #2, 18 Apr 2025**

Temperature index models can provide valuable estimates of snowpack characteristics and hydrological variables. One such model is examined in the present preprint. While there are compelling reasons to use simplified modelling, it must be acknowledged that state of the art snow modelling today can be much more complex, explicitly including multiple snow layers, snowpack energy balance, and more processes (wind redistribution, snowpack temperature gradients, liquid water in snowpack, refreeze). I found interesting ideas in this study, including the attempt to find a relationship for the spatial variability of the melt factor. However, I am concerned with the authors' claim to "comprehensively assess temperature index model assumptions, parameterizations, and performance across a range of snow climates". The study's scope is narrower than this and there are some issues that I believe need to be addressed.

We thank the reviewer for their thoughtful and constructive feedback. We appreciate that they found our study interesting and believe that their comments will help improve our manuscript.

We fully agree that state-of-the-art snow models based on energy balance formulations can resolve a much broader range of snowpack processes than the temperature-index approach. Our manuscript does acknowledge this (lines 50–58), but we will revise the introduction to more clearly and explicitly explain the limitations of temperature-index modelling relative to more complex physical models

However, the intent of our study is not to argue for the superiority of the temperature-index model, nor to suggest it as a substitute where full energy balance modelling is feasible. Rather, our goal is to provide a systematic, climate gradient evaluation of temperature-index model behaviour, assumptions, and performance. This is important given its widespread use, due partly to ease of implementation and minimal data requirements, especially where detailed forcing or energy balance data are unavailable. We will revise the abstract and discussion to better clarify this scope and intended contribution.

Regarding the concern that our claim to "comprehensively assess" TI model assumptions may overstate the scope, we appreciate the opportunity to clarify our intent. While we do not aim to evaluate every process or modelling approach in snow hydrology, we use the term 'comprehensively' in the context of TI modelling and its assumptions.

1. Suitability of the datasets. This SWE dataset, derived from snow depth observations and co-located with precipitation and temperature observations, covers extensive spatial and temporal domains. This is a great strength. However, the temperature index model linking precipitation, temperature, and SWE relies on a balance between accumulation and melt processes. Therefore, the fact that the observational datasets cannot detect when both accumulation and melt have happened within one time step is problematic. This is addressed as a limitation for snow decreases in L161, but the same problem could affect the assumption in L151. Findings presented in L198-203 could also be affected by days with both melt and accumulation occurring, in addition to the explanations offered by the authors. L210-220 suggest that some SWE changes are not detectable in this dataset, but once again this could be confounded by a mixture of processes occurring in one time step. While small SWE changes may indeed be missed in the dataset, this means there is added uncertainty on derived estimates such as onset of snow, peak SWE, onset of melt, etc. L244-246 highlight more errors implicit in using this dataset for this work. These issues could be examined by using some other dataset to examine the magnitude of errors introduced by some of the provided plausible explanations.

We thank the reviewer for these thoughtful points. As noted, the SWE dataset we use (NH-SWE), based on snow depth observations co-located with precipitation and temperature, is a major strength of this study due to its unprecedented spatial and temporal coverage. At the same time, we acknowledge that, like any observational dataset, it comes with limitations, many of which we have already discussed in the manuscript. We agree, however, that a few aspects of these limitations could be clarified or expanded, and we will address this in the revised version.

Regarding the concern about possible co-occurrence of accumulation and melt within a single daily time step, this is indeed a potential source of uncertainty, and we agree that we did not explicitly discuss it in sufficient detail. We will revise the manuscript to acknowledge this limitation. However, the available evidence suggests this issue is actually very rare and unlikely to bias our overall findings. More specifically:

- As shown in Figure 5c and discussed in lines 249–251, the potential for both melt and accumulation within a single day is mostly confined to temperatures near 0 °C. Days with recorded precipitation and SWE loss (indicative of possible rain-on-snow or mixed processes) represent fewer than 10% of events at 0 °C, and drop to less than 1% at −5 °C. y. **We will add discussion of this uncertainty in the revised manuscript.**

- Similarly, Figure 3c supports this interpretation as most cases with simultaneous precipitation and SWE loss above 0 °C are attributable to rain-on-snow rather than snowmelt–snowfall overlap.

- While we cannot resolve sub-daily variability with daily data, this limitation applies primarily to small-magnitude melt events during the accumulation season. Larger snowmelt events during the melt season are confidently captured, as

demonstrated by Fontrodona-Bach et al. (2023), which shows minimal bias in total melt estimates.

We also note that the statement in L151, that SWE increases can be confidently attributed to snow accumulation, is supported by the physical basis of this measurement. Increases in snow depth correspond to new snow accumulation, except in rare cases of measurement noise, which are negligible at the scale of this study.

Regarding the undetected SWE changes noted in L210–220 and L244–246, we confirm that these correspond to well-known issues such as snowfall undercatch or small precipitation events that fall below the detection threshold. Again, these represent a small fraction of the total dataset. Lines 216–220 and Figure 3d clarify that these are typically very small (<2 mm) and unlikely to significantly impact seasonal-scale results.

Furthermore, the NH-SWE dataset used here has been independently evaluated (Fontrodona-Bach et al., 2023), showing high accuracy in snow onset (forced by observations), and only small median biases in peak SWE (−1.7%) and melt onset (−1 day). These uncertainties are also very small relative to the interannual and spatial variability explored in our study.

In summary, whilst this dataset has some acknowledged limitations, including the inability to explicitly resolve overlapping processes within the daily time step, we are confident these do not materially impact the results and conclusions. We will however endeavour to clarify these points in the revised manuscript and strengthen our discussion of these specific uncertainties.

2. Structural uncertainty introduced by binary thresholds, some thresholds not varied. The snow accumulation and snow melt thresholds are both assumed to be strict cutoffs for the respective processes. However, observations (e.g. Dai, 2008) suggest that there is a smooth rain-snow phase transition, though much uncertainty and fundamental difficulties remain (e.g. Jennings et al., 2025) . What is the consequence of this structural assumption for the temperature index model? Furthermore, while 0C is a reasonable guess for a melt threshold, and the data does not contradict it, there is no test provided to show that it is the best choice and there is even some indication that another choice would be more suitable (as described in section 4.2, 5.1, and 5.3). Further tests could examine the melt threshold's effect on the relative performance of the three simulations.

We thank the reviewer for this insightful comment. We agree that the use of fixed temperature thresholds introduces structural simplifications, especially given evidence from higher-resolution studies (e.g., Dai, 2008; Jennings et al., 2025) that the rain–snow phase transition is more gradual. We will include these references in the revised introduction and clarify the implications of this simplification in our study.

**Snow accumulation threshold:**

We note that the above-cited studies use sub-daily (3-6 hourly) data, while our analysis is based on daily time steps. At daily resolutions, uncertainty in the timing of precipitation

events makes the application of these smoother temperature thresholds less straightforward. Nonetheless, we agree that this simplification deserves testing. We have therefore conducted additional sensitivity analyses using both fixed and smoothed snowfall temperature thresholds on our daily data.

To reduce confounding and noise from more minor accumulation events that have little impact on SWE, we focus on precipitation days with more than 10 mm recorded. As shown in Figure R1, no single fixed threshold performs optimally: low thresholds tend to misclassify snow days as rain, while higher thresholds misclassify rain as snow. Smooth thresholds, where snowfall probability transitions linearly between two temperatures, offer only marginal improvements for total snowfall but do not improve peak SWE or snow season onset (Figure R2).

[Figure]

Figure R1. Fraction of events with snow accumulation and no snow accumulation (rain). Based on 1,321,260 days with observed precipitation of more than 10 mm. If the precipitation caused an increase in snow accumulation, the precipitation is considered as snow.

Interestingly, a lower threshold (e.g., 0 °C) improves estimates of total snowfall and snow onset by reducing false accumulation on ephemeral snow days outside the core season (Figure R2 top panel). Conversely, a higher threshold (e.g., 1.5 °C) improves peak SWE by reclassifying marginal rain events as snow (Figure R2 middle panel). However, it is important to emphasise that our data clearly demonstrates that no threshold, fixed or smooth, consistently improves all metrics or reduces interquartile uncertainty ranges.

[Figure]

Figure R2. Sensitivity of results to varying fixed and smoothed snowfall temperature threshold. Black boxes indicate the results for fixed thresholds, and their neighbouring blue boxes show the results of a smooth snowfall threshold with a 1°C lower and 1°C higher range around the fixed threshold, where the amount of precipitation falling as snow linearly scales from all snow in the lowest temperature to all rain in the highest temperature. The grey shaded box indicates the threshold used in the original submitted manuscript.

**Snowmelt threshold:**

We also tested varying the melt temperature threshold (Figure R3). A slightly higher threshold (0.5 °C) does improve melt onset estimation, consistent with the physical requirement for additional energy input to initiate melting (Molotch et al., 2009; Jennings et al., 2018a). However, this same adjustment degrades performance in predicting snow season end dates, which are more sensitive to daily temperatures near 0 °C once melt has already begun.

As with snowfall thresholds, these results highlight trade-offs, whereby adjusting thresholds improves performance for some variables at the cost of others. Interquartile ranges remain broadly similar across threshold choices, suggesting that model performance is relatively insensitive to the precise value of the threshold within the ranges we tested.

[Figure]

Figure R3. Sensitivity of results to varying melt temperature thresholds. The grey shaded box indicates the threshold used in the original submitted manuscript.

Although the fixed thresholds used in our original analysis are simplifications, the new sensitivity tests show that alternatives yield mixed results and no consistent performance gains. We will include these new analyses as Supplement figures in the revised manuscript and expand our discussion of threshold-related structural uncertainty. We appreciate the reviewer's suggestion, which has helped strengthen the manuscript's methodological transparency.

3. Needs more focus on value added. There are interesting ideas in this paper, despite some data and methodological challenges.
(We split point number 3 in several subpoints).

Our manuscript investigates the performance, assumptions, and parameter sensitivities of the temperature-index (TI) modelling approach across a wide range of Northern Hemisphere snow climates. While our goal was not to develop or optimize a new TI model, our analysis offers several novel contributions: 1) a systematic cross-climatic evaluation of TI model behaviour across >5,000 sites, 2) a transparent assessment of performance limitations under different snowpack regimes, 3) a sensitivity analysis of

model assumptions (e.g. thresholds, melt factors) and their spatial variability, 4) new spatial analyses that reveal how and where model performance varies systematically with snow climates. We recognize that the discussion and conclusion did not highlight these contributions strongly enough, and will ]revise these accordingly.

I believe that refocusing on the regional results is more critical than summary statistics covering the large regions. We see in Figure 10 that many regions remain poorly represented (description L314-318), but the reasons are left mostly unexplored.

We agree that further exploration of regional performance would strengthen the manuscript. While the smaller scale variability is outside the scope of this work, we have conducted additional analyses to clarify regional patterns in model error. For example, Figure R4 shows spatial patterns of over and underestimation in peak SWE. Interestingly, the errors differ by snow regime and can be divided into 1) continental areas with shallow snowpacks, where peak SWE is often overestimated, likely due to unaccounted sublimation or wind redistribution, and 2) temperate, higher elevation regions with deeper snowpacks, where undercatch of solid precipitation may explain the underestimation bias (see also Cho et al., 2022). We will include similar spatial analyses for other key metrics in the revised manuscript and explicitly interpret these findings in the context of climate and measurement uncertainty.

[Figure]

Figure R4. Spatial distribution of peak SWE over-/under- estimation.

Further, while the median performance for most snow metrics is good (Figure 9), the box and whiskers show a large range. Which regions specifically contribute to this wide spread?

We agree that we can better explain the wide range of errors observed in Figure 9. As shown in Figure R5, relative errors in peak SWE are largest at stations with very shallow

snowpacks (<100 mm). For such sites, small absolute deviations translate to large relative errors (e.g. 100 mm modelled vs 10 mm observed = +1000%). However, for stations with deeper snowpacks, errors are consistently lower. Median underestimation for snowpacks >100 mm is ~25%. We will include this analysis and complementary figures in the revised manuscript. It will help clarify that while outliers are prominent, they are concentrated in a specific subset of sites and not indicative of general performance.

[Figure]

Figure R5 . Analysis of relative errors of peak SWE. All station-years (left) and median per station (right).

How do all of these results comparable to observational uncertainty or model uncertainty from state-of-the-art snow models? Re-focusing the study on new findings and placing the results in context is needed to justify the conclusion that the model performs satisfactorily.

This is an important contextual point. While a direct comparison to energy balance models is not possible at this scale, due to the lack of input data and information on appropriate parameter ranges, we can offer a useful benchmark by comparing reported performance from recent model evaluations. Menard et al. (2021) report peak SWE biases from −50% to +35% across ESM-SnowMIP models. Cho et al. (2022) found that land surface models underestimated peak SWE by 268 mm on average at SNOTEL sites. In comparison, our TI-based approach underestimates peak SWE at SNOTEL sites by 114 mm on average, and the timing of peak SWE is 23 days early vs. 36 days early in Cho et al. (2022).
We will incorporate this comparison into the discussion and include a new boxplot showing site-median performance (currently only mapped in Figure 10). While such comparisons are not one-to-one, they demonstrate that TI models, when interpreted cautiously, can yield useful constraints.

While particularly intriguing because it could offer a way to extrapolate melt factors from the point-scale, the multi-linear regression model for melt factor also appears minimally predictive. In addition to the good practice of preserving a random subset of data from

the regression in order to use it for testing, other variables could be added (longitude, mean SWE amount, perhaps a typical "snow type" (Sturm & Liston, 2021).

We agree that the predictive power of the regression model is modest. As suggested by both reviewers, we will withhold a random test set to validate the model and assess generalisability more robustly. We have already tested a range of covariates, but we will additionally explore whether including a snow type classification (e.g. Sturm & Liston, 2021) improves predictive skill. We will include the updated regression analysis and testing results in the revised manuscript.

Other questions and minor points:

L62 typo "toe" should be "to"

Will correct.

L112-115/L445-453: How much missing data is there that needs gap-filling? Could this affect any of the results (e.g. giving deltaSWE=0)?

We will provide numbers for the gap filling.

L170: It could be possible for melt to occur on days with negative temperatures (e.g. if some period of the day exceeded 0C). Could it also be explained by the snow depth decreasing due to wind compaction?

The ΔSNOW model (Winkler et al. 2021) is used to convert snow depth to SWE and accounts for this process, but it is possible that in some cases the model reaches the maximum density (which is parameterised) and sees a further decrease in snow depth as melt, when it actually was further compaction. A more comprehensive analysis is provided by Winkler et al. (2021) and Fontrodona-Bach et al. (2023). We will add this possibility (model inaccuracy) in the specified line.

L235: Would equation (5) be sensitive to changes in the binary structure of equation (3)?

Equations 3 and 5 are independent and do not interact with each other. While equation 3 describes the snow accumulation routine in the temperature-index approach, equation 5 describes a model that derives the accumulation temperature threshold from the NH-SWE time series and the observations of temperature, but there is no modelling involved.

L242 typo "is only be" should be "is only"

Will correct.

L250 typo "blue and read" should be "blue and red"

Will correct.

L260-261, L375-376: It would be good to really highlight these spatial results which show where temperature index models can or can't work robustly.

This is shown in Figure 8b but is perhaps hidden due to overlapping of many points. We will make this more explicit. This will anyway be more thoroughly discussed also in reply to main comment number 3.

L386-388: It is somewhat misleading to present hemispheric scale results when there are such big regional differences. Could focus the summary text onto the range of results in Figure 10, perhaps then aggregating results (e.g. for Western North America, Arctic, Europe, and Scandinavia) to give median summary statistics with regional grouping.

We agree and will incorporate this suggestion also in reply to main comment number 3.

L404-407: I think this is a key aspect for discussion. Should be revisited in the Conclusions section.

Yes we will revisit this, we agree it is an interesting finding. In response to main comment number 3, we will also provide a more detailed analysis of the errors in melt rates.

L416-418: Could reference other data products that use simple snow modelling and data assimilation of snow observations already (e.g. CMC daily snow depth product; Brown and Brasnett, 2010), or else clarify if "assimilation" used here means something different.

As we do not really test the possibility of snow data assimilation, we think it might be best to remove this statement.

L420: Could the SWE decreases below freezing be due to some other explanation, including observational errors? If such errors are potentially present, then could they affect your other findings?

They could also be observational errors, similar to the response to the minor comment about line L170. This is discussed in more detail in reply to major comment number 1.

L419-423: There were no tests showing 0C to be more valid than another choice of melt threshold, or another method of partitioning precipitation into rain and snow. It was fixed throughout the whole study.

These tests are now provided. This is discussed in more detail in reply to major comment number 2.

L424-427: How do these melt factors compare to those found in the literature? With respect to the values of the two rates, this is still a large range. While it has been diagnosed that there are regions where there is high interannual variability in melt

factors, what might cause this variability? Does it disqualify temperature index modelling from being used in those regions?

In the revised manuscript we will contextualise the values of melt factors that our study finds. We will also discuss potential reasons for the high interannual variability. We do not think it disqualifies TI modelling in those regions, since the performance evaluation of melt rates is good, but we will delve into the reasons why the performance is good despite the high interannual variability.

L429: Modelling peak SWE requires the right balance of accumulation and ablation processes. Which of these is most contributing to these challenges?

This is an interesting point. We will include an analysis of this in our revised manuscript.

NEW REFERENCES:

Cho, E., Vuyovich, C. M., Kumar, S. V., Wrzesien, M. L., Kim, R. S., & Jacobs, J. M. (2022). Precipitation biases and snow physics limitations drive the uncertainties in macroscale modeled snow water equivalent. Hydrology and Earth System Sciences, 26(22), 5721–5735. https://doi.org/10.5194/hess-26-5721-2022

Krinner, G., Derksen, C., Essery, R., Flanner, M., Hagemann, S., Clark, M., Hall, A., Rott, H., Brutel-Vuilmet, C., Kim, H., Ménard, C. B., Mudryk, L., Thackeray, C., Wang, L., Arduini, G., Balsamo, G., Bartlett, P., Boike, J., Boone, A., … Zhu, D. (2018). ESM-SnowMIP: Assessing snow models and quantifying snow-related climate feedbacks. Geoscientific Model Development, 11(12), 5027–5049. https://doi.org/10.5194/gmd-11-5027-2018